# Explainable Zero-Shot Visual Question Answering via Logic-Based Reasoning

**Thomas Eiter**                                                THOMAS.EITER@TUWIEN.AC.AT
**Jan Hadl**                                                    JAN.HADL@TUWIEN.AC.AT
**Nelson Higuera**                                             NELSON.RUIZ@TUWIEN.AC.AT
*Vienna University of Technology (TU Wien), Austria*

**Lukas Lange**                                                LUKAS.LANGE@DE.BOSCH.COM
*Bosch Center for Artificial Intellgince, Renningen, Germany*

**Johannes Oetsch**                                            JOHANNES.OETSCH@JU.SE
*Jönköping University, Sweden*

**Bileam Scheuvens**                    BILEAM.SCHEUVENS@STUDENT.UNI-TUEBINGEN.DE
*University of Tübingen, Germany*

**Jannik Strötgen**                                            JANNIK.STROETGEN@H-KA.DE
*Karlsruhe University of Applied Sciences, Germany*

**Editors:** Leilani H. Gilpin, Eleonora Giunchiglia, Pascal Hitzler, and Emile van Krieken

## Abstract

Visual Question Answering (VQA) is the task of answering natural language questions about images, which is a challenge for AI systems. To enhance adaptability and reduce training overhead, we address VQA in a zero-shot setting by leveraging pre-trained neural modules without additional fine-tuning. Our proposed hybrid neurosymbolic framework, whose capabilities are demonstrated on the challenging GQA dataset, integrates neural and symbolic components through logic-based reasoning via Answer-Set Programming. Specifically, our pipeline employs large language models for semantic parsing of input questions, followed by the generation of a scene graph that captures relevant visual content. Interpretable rules then operate on the symbolic representations of both the question and the scene graph to derive an answer. Our framework provides a key advantage: it enables full transparency into the reasoning process. Using an existing explanation tool, we illustrate how our method fosters trust by making decisions interpretable and facilitates error analysis when predictions are incorrect. Beyond explaining its own reasoning, our framework can also explain answers from more opaque models by integrating their answers into our system, enabling broader interpretability in VQA.[1]

## 1. Introduction

AI systems capable of answering questions about images are increasingly vital across various applications (Barra et al., 2021; Lin et al., 2023). This task, known as *Visual Question Answering* (VQA) (Antol et al., 2015; Goyal et al., 2017), requires a joint understanding of visual and textual modalities and the ability to follow reasoning steps—whether implicitly or explicitly—to arrive at the correct answer. Beyond accuracy, explainability is crucial for fostering trust and facilitating error analysis, a challenge that is particularly pronounced in end-to-end deep learning systems.

Neurosymbolic approaches to VQA offer a promising solution to the interpretability challenge. They employ neural models to translate multi-modal inputs into symbolic representations and then

---

1. Code and data are available from https://github.com/pudumagico/nesy25.

use symbolic execution to derive answers (Yi et al., 2018; Mao et al., 2019; Amizadeh et al., 2020; Eiter et al., 2022; Surís et al., 2023; Johnston et al., 2023; Abraham et al., 2024). This compositional architecture not only enhances transparency but also enables the easy exchange of components when improved alternatives emerge. Early work implemented the reasoning component in Python (Yi et al., 2018), but recent efforts (Abraham et al., 2024; Eiter et al., 2023, 2022; Basu et al., 2020) favor logic-based approaches, particularly *Answer-Set Programming* (ASP) (Brewka et al., 2011; Lifschitz, 2019), which allows reasoning steps to be declaratively specified with rule-based models. Prior studies have shown that ASP can resolve ambiguities in neural outputs (Eiter et al., 2022) and can also provide diagnostic reasoning to explain answers (Eiter et al., 2023). However, these works rely on abridged pipelines that omit question parsing by assuming a pre-constructed symbolic representation of reasoning steps, and they were evaluated on synthetic datasets (Johnson et al., 2017) that do not capture the complexity of real-world scenarios.

In this work, we tackle VQA in a *zero-shot setting*[2] by leveraging pre-trained neural models without additional fine-tuning, thereby enhancing adaptability and reducing training overhead. We introduce a hybrid neurosymbolic framework that integrates neural and symbolic components via logic-based reasoning using ASP, leveraging existing work. Unlike prior ASP-based VQA approaches that rely on predefined symbolic representations, our method processes raw natural language questions by integrating LLM-based parsing, which ensures broader applicability to real-world datasets. As a testbed for our pipeline, we use the challenging GQA dataset (Hudson and Manning, 2019) that features real-world images with complex scenes and diverse questions with many possible answers. Our approach employs a pipeline, illustrated in Figure 1, that consists of the following key steps:

1. *Language Module:* Semantic parsing of input questions using a large language model (LLM) into a symbolic representation;
2. *Vision Module:* Generating a scene graph by using a novel method to focus only on the relevant visual content; and
3. *Reasoning Module:* Deriving an answer by using interpretable ASP rules with the symbolic question representation and the scene graph with an ASP solver.

Our framework provides a key advantage over other VQA approaches: it enables full transparency in the reasoning process. We demonstrate how the explanation tool xclingo (Cabalar et al., 2020) enhances transparency by tracing the reasoning process, thereby facilitating error analysis of incorrect predictions. Beyond explaining its own reasoning, our framework can also explain answers from more opaque models by integrating their answers, thus enabling broader interpretability in VQA.

In Section 2, we consider related work while in Section 3, we review technical background on ASP. In Section 4, we detail our end-to-end pipeline for processing GQA problems that combines LLM-based semantic question parsing, question-centric scene graph generation, and ASP for reasoning, and present an experimental evaluation against related approaches. Section 5 shows how xclingo can be used to explain answers and localise errors of incorrect predictions. Section 6 concludes the paper.

## 2. Related Work

We take advantage of prior work (Eiter et al., 2022) that applied ASP to VQA on the CLEVR dataset (Johnson et al., 2017). However, CLEVR consists of simple scenes with few objects and does not require complex scene graphs. Additionally, Eiter et al. (2022) assumed pre-constructed

---

2. We use "zero-shot" to refer to a *no-learning setting*, where no training or fine-tuning is performed, though in-context examples may be used.

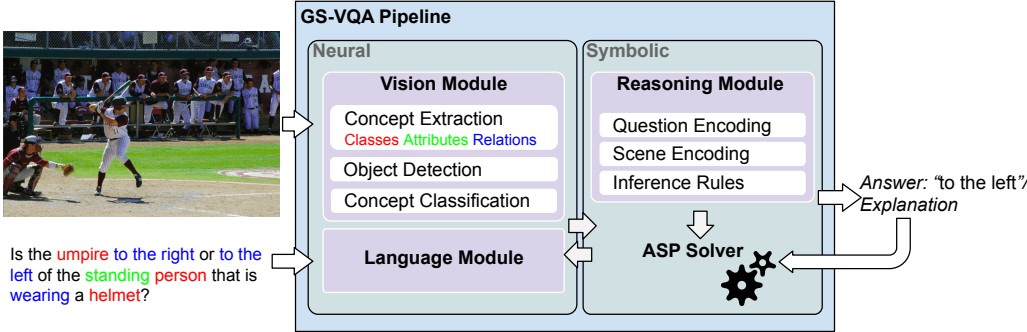

Figure 1: Overview of our Grounded-Scene Visual Question Answering (GS-VQA) pipeline.

symbolic representations of questions, omitting the parsing step. We address these limitations by extending ASP-based reasoning to the GQA dataset and integrating LLM-based question parsing.

While we use a fixed ASP theory, recent work has explored more adaptive approaches. Eiter et al. (2024) proposed learning new ASP rules from unseen examples, and Chowdhury et al. (2025) developed a framework for dynamically acquiring domain knowledge. These advances complement our work by pointing toward more flexible symbolic reasoning.

Like several recent approaches to GQA, our method operates in a zero-shot setting to avoid task-specific fine-tuning. Standalone vision-language models (VLMs) such as BLIP-2 (Li et al., 2023) were shown to generalize well across multiple datasets, including GQA. Modular approaches like ViperGPT (Surís et al., 2023), CodeVQA (Subramanian et al., 2023), and PnP-VQA (Tiong et al., 2022) integrate these models as components and achieve better performance. However, these methods rely on VLMs to handle entire reasoning steps in an opaque manner, which limits interpretability. In contrast, our ASP-based pipeline remains transparent and explainable.

Using LLMs to convert natural language questions into ASP fact representations has been explored by Bauer et al. (2025) in the context of VQA on images of graphs. However, the questions in that domain were rather simple and could be parsed using regular expressions, unlike the more complex queries in GQA. Leveraging LLMs to extract logical predicates for ASP-based reasoning has also been investigated beyond VQA, as Rajasekharan et al. (2023) did e.g. for qualitative reasoning, mathematical reasoning, and goal-directed conversation. Beyond VQA, ASP has been successfully applied to various neurosymbolic tasks, including image segmentation (Bruno et al., 2021), processing sequences of sensory input (Evans et al., 2021), and visual validation of control panels (Barbara et al., 2023).

Explainability is crucial for building transparent and trustworthy VQA systems (Dosilovic et al., 2018). Some approaches focus on visualising pixel contributions to predictions (Arras et al., 2022), yet such techniques provide only coarse insights into the underlying reasoning. For instance, the end-to-end MAC system (Hudson and Manning, 2018) leverages attention mechanisms to highlight relevant image areas and question elements, but its visual cues remain superficial and do not reveal detailed reasoning steps. While modular VQA architectures like NSVQA (Yi et al., 2018) improve interpretability by disentangling perception from reasoning, logic-based methods offer additional flexibility by supporting both deductive (forward-oriented) and diagnostic (backward-oriented, abductive) reasoning. Recent work (Eiter et al., 2023) employed ASP to generate contrastive

explanations (Lipton, 1990) for CLEVR—clarifying why one answer is chosen over another. In contrast, our approach uses an existing ASP explanation tool (Cabalar et al., 2020) to directly trace the reasoning process, providing a clear and fine-grained account of how answers are derived.

## 3. Answer-Set Programming: Preliminaries

Answer-Set Programming (ASP) (Brewka et al., 2011; Lifschitz, 2019) is a declarative problem-solving framework rooted in logic programming. It allows users to specify a problem using logical rules, and efficient solvers automatically compute solutions that satisfy these rules. We use the clingo solver (v. 5.6.2) from the Potassco toolkit.[3]

At its core, an ASP *program* $P$ consists of a finite set of *rules* of the form

$$a :- \ b_1, \ldots, \ b_m, \ not \ c_1, \ldots, \ not \ c_n \quad m, n \geq 0,$$

where $a$, and all $b_i$, $c_j$ are logical atoms in a first-order language (basic facts that can be true or false), and $not$ stands for negation as failure. The informal reading is "If all $b_i$ are true and no $c_j$ is true, then $a$ must be true." If a rule has no head ($a$ is missing), it acts as a *constraint*, eliminating solutions that violate the condition. A *fact* is a rule with $m = n = 0$ stating that $a$ is unconditionally true.

The semantics of a variable-free ASP program is given by *answer sets*, which are models that satisfy a stability condition (Gelfond and Lifschitz, 1988). A (Herbrand) interpretation of $P$ is a set $I$ of ground atoms in the language of $P$ (intuitively, the atoms that are true); $I$ is a model of $P$ if for each rule $r \in P$ either $(i)$ $a \in I$ or $(ii)$ $\{b_1, \ldots, b_m\} \not\subseteq I$ or $(iii)$ $I \cap \{c_1, \ldots, c_n\} \neq \emptyset$ holds; i.e., $I$ satisfies $r$ viewed as implication in classical logic. Furthermore, $I$ is an answer set of $P$, if $I$ is a $\subseteq$-minimal model of the program $P^I = \{r \in P \mid \{b_1, \ldots, b_m\} \subseteq I \text{ and } I \cap \{c_1, \ldots, c_n\} = \emptyset\}$. Intuitively, $I$ must result by applying the rules $r$ whose bodies "fire" w.r.t. $I$ starting from facts.

For programs containing variables, ASP solvers automatically ground them—replacing variables with all possible values—before computing answer sets.

Further ASP features are *choice rules* and *weak constraints*. Choice rules allow multiple possible solutions while enforcing constraints on the number of selected elements. Weak constraints introduce preferences, enabling optimization by minimizing costs. These are particularly useful for handling uncertainty or selecting the most likely solution among alternatives. For further details, the reader is encouraged to consult the works of Brewka et al. (2011) and Calimeri et al. (2020).

## 4. The GS-VQA Pipeline

In this section, we present the Grounded-Scene Visual Question Answering (GS-VQA) pipeline—a comprehensive framework that integrates LLMs, Vision-Language Models (VLMs), and ASP—to address VQA in a transparent, zero-shot setting. We first introduce the GQA dataset and then give an overview of the pipeline; further technical details are available in the project's online repository.

### 4.1. The GQA Dataset

We use the GQA dataset, a benchmark that has been extensively used in recent work (Amizadeh et al., 2020; Surís et al., 2023; Liang et al., 2020; Li et al., 2023). It contains over 22 million questions, including both open-ended and binary formats, which exhibit complex structures, require diverse reasoning skills, and span a large set of $1\,878$ possible answers. The questions are grounded in more

---

3. https://potassco.org/

than 100 000 images from the Visual Genome dataset (Krishna et al., 2017), which depict real-world scenes with a diverse range of object classes, attributes, and relationships.

GQA further provides two types of supplementary data: (1) every natural-language question has an associated *functional program*, which is a structured representation outlining the required reasoning steps, and (2) every image comes with a Visual Genome *scene graph*, which allows us to assess the soundness of reasoning under idealised, noise-free visual input.

## 4.2. Overview of our VQA Pipeline

To solve GQA, we adhere to a modular neurosymbolic architecture of VQA systems, illustrated in Figure 1, that has the following core components:

- **Language module:** Neural networks that process the natural-language question to produce a symbolic representation of the question.
- **Vision module:** Neural networks that process the visual scene to produce a symbolic representation of the scene. Notably, we use *"question-driven" partial scene-graph extraction*, where only information is extracted from the scene that is relevant for answering the question at hand.
- **Reasoning module:** A component that uses an ASP solver to derive the answer. The solver takes a scene and a question representation in the form of ASP facts as input, and ASP rules encode the reasoning steps to derive the answer.

For all the neural components, we use pre-trained models.

### 4.2.1. LANGUAGE MODULE WITH LLMS FOR QUESTION PARSING

The language component converts GQA questions into structured representations that downstream modules can process. To achieve this, we employ transformer-based LLMs (Vaswani et al., 2017), such as GPT-4 (OpenAI, 2024), which have demonstrated remarkable performance on similar tasks. These models require no task-specific training; instead, they leverage in-context learning, where examples of the task are provided as part of the input prompt.

For conversation-based models, a preprompt defines the LLM's role as a *system prompt*, which is followed by in-context examples and the question to be translated as a *user prompt*.

The system prompt instructs the model to adopt the persona of a question translator and provides general guidelines:

```
You are now a question parser. Your task is to translate a question
into a functional program. The available operations are: select, relate,
common, verify, choose, filter, query, same, different, and, or, exist.
```

The user prompt is a template that includes sample questions and their corresponding translations, followed by a directive to translate the next question, and finally the question itself. Unlike the system prompt, the user prompt is populated at runtime with the actual question and examples. For instance, a filled-in user prompt might look like this:

```
Here are examples of questions and corresponding programs:
What is the dish inside of?
query(unique(relate_any(select(scene(), dish), inside, object)), name)
Can you translate the following question into an ASP program?
Is there a green car?
```

Given the limited context window of LLMs and that LLMs capable of large context windows struggle with too many examples (Liu et al., 2023), in-context examples must be carefully selected to maximise the information conveyed. We balance the need for examples similar to the target question with the requirement for diversity. While some approaches—such as the one by Subramanian et al. (2023)—rely solely on similarity, we combine both strategies. First, we gather examples that closely match the target question; then, we add examples featuring underrepresented operations until all functional program operations are covered. Cosine similarity of BERT embeddings (Devlin et al., 2019) is employed as our measure, and candidate examples are drawn from the GQA training set.

Finally, the functional representations of questions are translated into ASP facts. For example, the question "Is the umpire to the right or to the left of the standing person that is wearing a helmet?" is rendered as follows:

```
scene(0). select(1, 0, helmet).
relate(2, 1, person, wearing, subject).
filter_any(3, 2, standing).
choose_rel(4, 3, umpire, to_the_left_of, to_the_right_of, subject).
end(4).
```

Here, the numbers represent steps in the evaluation plan: the first argument indicates the current step, while subsequent numbers refer to previous steps that provide the necessary input.

### 4.2.2. VISION MODULE WITH QUESTION-DRIVEN SCENE GRAPH GENERATION

While several zero-shot scene graph generation approaches have been proposed (Chang et al., 2023), we develop a tailored method for better alignment of the concepts in the question and ASP rules with the scene. Moreover, generating a full scene graph—where every detected object is annotated with likelihoods for all possible classes, attributes, and relations—is impractical for the visually complex scenes in GQA. To address this challenge, the pipeline employs *question-driven partial scene-graph extraction*, which extracts only the information relevant to answering the question.

To this end, the *concept-extraction* component identifies the pertinent object classes, attributes, and relations from the semantic representation of the input question. Essentially, it processes the question in its functional form to produce a tuple $(C, A, R)$, where $C$ is a set of classes, $A$ is a set of attribute categories, and $R$ is a set of relations. For the example question in Figure 1, the tuple is

$$(\{\texttt{helmet}, \texttt{umpire}, \texttt{person}\}, \{\texttt{pose}\}, \{\texttt{wearing}, \texttt{to\_the\_left\_of}, \texttt{to\_the\_right\_of}\}).$$

Utilising the $(C, A, R)$ tuple from the concept-extraction component, the *scene-processing component* is responsible for extracting a question-driven partial scene graph. This graph is represented as a list $O = [o_1, \ldots, o_n]$ of objects where each object $o_i = (id, s, B, c, A_o, R_o)$ comprises: a unique identifier $id$, a confidence score $s$ (ranging from 0 to 1) indicating the certainty of the object detection, a bounding box $B$, a class $c$, and sets $A_o$ and $R_o$ containing attribute and relation likelihoods.

The class $c$ is either an element of $C$ or a most specific subclass of some $c' \in C$. This ensures that objects are identified with precise classifications (e.g., "baseball player" instead of just "person"), resulting in a scene graph that includes only objects pertinent to the question. For each attribute category $a \in A$ and possible value $v$ for $a$, $A_o$ provides a likelihood (between 0 and 1) that $v$ applies to the object. Similarly, for each relation $r \in R$ and distinct detected objects $o_i \neq o_j$, $R_o$ contains a likelihood (between 0 and 1) that $r(o_i, o_j)$ holds.

We use OWL-ViT (Minderer et al., 2022) for *object detection*, i.e., for identifying relevant objects and their bounding boxes in the scene, and the foundation model CLIP (Radford et al., 2021) for *concept classification*, i.e., which queries the image for attributes and relations of the detected objects.

### 4.2.3. REASONING MODULE WITH ANSWER-SET PROGRAMMING

Utilising ASP for symbolic reasoning offers a robust framework equipped with mature tools and solvers. More importantly, ASP enables the incorporation of uncertainty inherent in the outputs of the image-processing component.

The scene graph $O = [o_1, o_2, \ldots, o_n]$ from scene processing is translated into our ASP representation by converting each object with its associated attributes and relations into ASP facts. In the translation, likelihoods are respected. For a scene containing an object whose most specific class is "baseball_player" as in Figure 1), the corresponding ASP facts might be:

```
object(o1). has_obj_weight(o1, 1971).
has_attr(o1, class, alive).  has_attr(o1, class, person).
has_attr(o1, class, baseball_player).
has_attr(o1, name, baseball_player).
{ has_attr(o1, pose, standing) }.
:~ has_attr(o1, pose, standing).     [83,   (o1, pose, standing)]
:~ not has_attr(o1, pose, standing). [2525, (o1, pose, standing)]
```

The fact `object(o1)` declares the existence of an object `o1`. The `has_attr/3` predicates associate `o1` with various attributes, such as its class and name. The choice rule that follows allows the solver to consider the possibility that `o1` is standing. The subsequent weak constraints penalise the inclusion or exclusion of this attribute based on confidence scores, guiding the solver towards the most probable interpretation.

The semantics of the reasoning operations are defined using ASP rules in a uniform manner. Unlike the question and scene encodings, these rules are fixed and do not change from one question to another. This ASP program is solved alongside both scene and question encodings to produce an answer. The complete theory, available in the online repository, has 60 rules; here is one example:

```
state(TO,ID) :- filter_any(TO, TI, VALUE), state(TI, ID),
                has_attr(ID, ATTR, VALUE).
```

In this rule, `TI` and `TO` are variables representing input/output step references, and `ID` represents an object identifier. The rule filters all objects with a given value for any attribute.

### 4.3. Evaluation

We conducted a series of experiments on the GQA dataset to evaluate the feasibility of our approach. Our main objectives were: (1) to compare GS-VQA's performance against other zero-shot methods and (2) to assess the contribution of its individual modules to overall performance.

**Evaluation of the language model.**   We first evaluate the language component in isolation. GQA provides a scene graph for each image and a functional program for each question. Accuracy is measured by executing our generated question representation and comparing the output to the reference functional program on the scene graph. To account for linguistic variation, we use WordNet (Miller, 1995) to match close synonyms. For the language parser, we selected OpenAI's gpt-4o LLM (OpenAI, 2024), which outperformed several alternatives (a detailed comparison is

| Model | CodeVQA | ViperGPT | PnP-VQA | BLIP-2 | FewVLM | GS-VQA (ours) |
|---|---|---|---|---|---|---|
| Accuracy | 49.0% | 48.1% | 42.3% | 44.7% | 29.3% | 36.2% |

Table 1: GS-VQA's accuracy against other zero-shot approaches for GQA.

provided as part of the online repository). With between 12 and 18 in-context examples (10 of them selected using BERT similarity), such that every logical operation is present. The language component achieves an accuracy of 84.4%.

**Evaluation of the full pipeline.** We evaluated the full GS-VQA pipeline on a representative sample of 500 questions from GQA's balanced test-dev set, consisting of 12 578 questions. For object detection, we used the ViT-L/14 variant of OWL-ViT (Minderer et al., 2023), while concept classification was performed with the ViT-B/32 variant of CLIP (Radford et al., 2021).

We compared GS-VQA against various zero-shot systems: CodeVQA (Subramanian et al., 2023) and ViperGPT (Surís et al., 2023), which extract a symbolic representation from the question alone; PnP-VQA (Tiong et al., 2022), which derives a symbolic representation of the image but does not perform fully symbolic reasoning; and two end-to-end models, BLIP-2 (Li et al., 2023) and FewVLM (Jin et al., 2022), which rely entirely on neural networks for answer generation. To ensure a fair comparison, WordNet was not used in this evaluation.

The results are summarised in Table 1. GS-VQA correctly answers 36.2% of questions in GQA's test-dev set, while the best-performing zero-shot model, CodeVQA (Subramanian et al., 2023), achieves 49.0%, followed closely by ViperGPT (Surís et al., 2023). However, since both CodeVQA and ViperGPT generate Python code that may call another VQA model, their performance is inherently dependent on their respective backends—PnP-VQA (Tiong et al., 2022) and BLIP-2 (Li et al., 2023)—which serve as their effective baselines.

Unlike other approaches, ours decomposes question answering into fine-grained reasoning operations, enabling transparent execution and offering a clear advantage in interpretability. However, this comes at the cost of GS-VQA's accuracy currently lagging behind the leading zero-shot approaches on GQA. Given the modular nature of our architecture, individual components can be easily replaced with improved versions, allowing performance to scale as foundational models advance. Moreover, GS-VQA offers a unique advantage: it can not only explain its own reasoning but also be combined with stronger, potentially more opaque base models to provide explanations for their answers in terms of the given scene graph. This enables both higher accuracy from the base model and enhanced explainability through GS-VQA—a synergy we explore further in the next section.

## 5. Explaining Answers

Since GS-VQA orchestrates primitive operations through intuitive and compact ASP rules, it inherently offers a high degree of transparency. This rule-based approach also enhances maintainability, making it easier to adapt the pipeline to different tasks. To further clarify why a particular answer was produced, the ASP community has developed several notions and methods for *explainability* (Fandinno and Schulz, 2019; Oetsch et al., 2018; Schulz and Toni, 2016; Fandinno, 2016; Viegas Damásio et al., 2013; Oetsch et al., 2010; Pontelli et al., 2009), ranging from derivation tracing to alternative notions with input from social sciences (Miller, 2019), which were adopted e.g. by Eiter et al. (2023). We instead focus on a tracing-based explanation strategy.

```
|__The answer to the question is train_car
|  |__Between name carriage or train_car, we choose train_car
|  |  |__ The object with ID o3 is the only object detected
|  |  |  | of this kind
|  |  |__Object with ID o0 is in object with ID o3
|  |  |  |__ We select object o0 because of being in the
|  |  |  |  | class man
|  |  |  |  |__We locate a candidate from the detected objects,
|  |  |  |  |  object with ID o0
```

Figure 2: Derivation tree for the question "Is the man in a carriage or a train car?".

## 5.1. Tracing ASP derivations

To trace GS-VQA's reasoning process, we use the dedicated tool xclingo (Cabalar et al., 2020). This tool processes ASP programs that have been manually extended with markup annotations in the form of comments specifying which atoms or rules to explain. In particular, such an annotation specifies which text should be produced when the rule "fires" in a derivation. The ASP solver then uses these annotations to generate derivation trees with textual justifications, allowing for a step-by-step tracing of how an answer was derived.

For example, the following rule filters objects based on a fixed attribute value and includes an annotation for explanation:

```
%!trace_rule {"Selected object % with value % for attr. %.", ID, V, A}
state(TO,ID) :- filter(TO, TI, A, V), state(TI, ID), has_attr(ID, A, V).
```

If this rule is used in a derivation, the placeholders (%) are replaced by variable values from the rule instance, producing a natural-language explanation. Such annotations are easy to write and add additional documentation; the complete annotated rule set is given in the online repository.

A complete xclingo derivation tree for the question "Is the man in a carriage or a train car?" is given in Figure 2. This explanation tree helps users understand the reasoning steps taken by GS-VQA, allowing them to locate referenced objects in the image and follow the decision-making process step by step. Starting from the bottom, the trace first identifies an object as a man, which can be verified by inspecting the bounding box containing this object. Next, the trace establishes that the man satisfies the "is-inside-of" relation with a unique object. To determine the answer, the system evaluates the attribute associated with this object, considering two possible values: "carriage" and "train_car", ultimately selecting the latter as the correct response.

## 5.2. Localising errors

While explanation features are useful to foster trust when predictions are correct, they can be even more valuable when predictions are incorrect, helping to diagnose errors. There are two principal sources for mistakes in GS-VQA: the language component and the vision component. Errors in the language component are straightforward to detect, as we can directly compare the generated symbolic representation of reasoning steps with the original natural language question.

In contrast, errors in the vision module are harder to pinpoint. Possible issues are that relevant objects are not detected, or that objects, their attributes, or relations among them are misclassified. All types of errors can be identified using derivation traces. We present an example of a question that has been answered incorrectly due to a misclassified attribute in Figure 3. When asked "What is the person in front of?", we expect "sky", but GS-VQA incorrectly responds with "glove". Following the tree shows that the wrong object has been selected when resolving the in-font-of relation.

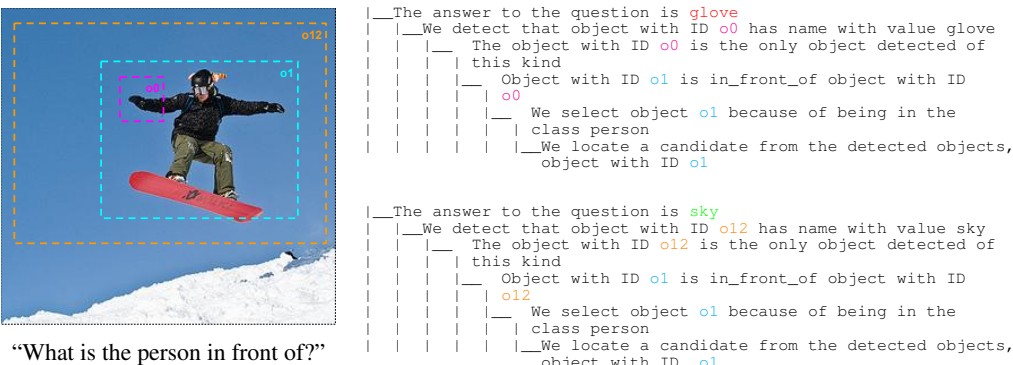

```
|__The answer to the question is glove
|  |__We detect that object with ID o0 has name with value glove
|  |  |__  The object with ID o0 is the only object detected of
|  |  |  | this kind
|  |  |  |__  Object with ID o1 is in_front_of object with ID
|  |  |  |  | o0
|  |  |  |  |__  We select object o1 because of being in the
|  |  |  |  |  | class person
|  |  |  |  |  |__We locate a candidate from the detected objects,
|                    object with ID o1

|__The answer to the question is sky
|  |__We detect that object with ID o12 has name with value sky
|  |  |__  The object with ID o12 is the only object detected of
|  |  |  | this kind
|  |  |  |__  Object with ID o1 is in_front_of object with ID
|  |  |  |  | o12
|  |  |  |  |__  We select object o1 because of being in the
|  |  |  |  |  | class person
|  |  |  |  |  |__We locate a candidate from the detected objects,
|                    object with ID  o1
```

"What is the person in front of?"

Figure 3: The derivation tree at the top is produced for the incorrect answer "glove", while the one at the bottom is obtained when enforcing the correct answer "sky" using a constraint.

### 5.3. Explaining answers of other VQA models

A distinguishing feature of our system is its ability to provide explanations for the answers of other models that may achieve higher accuracy but lack transparency. For instance, a model like Blip-2, which shows slightly better accuracy on the GQA dataset, might correctly predict the answer "sky" to the question "What is the person in front of?" about the image in Figure 3. However, as an end-to-end trained system, it does not provide insights into how this answer was derived.

To address this, we can add the constraint ":- not ans(sky)." to enforce the answer in GS-VQA, which modifies the reasoning process, adjusting the selection of objects, attributes, and relations to align with the given answer (if possible). As a result, we obtain a corresponding derivation tree, shown in Figure 3, relative to our scene graph. This allows us to provide an interpretable explanation even when the answer comes from a model that inherently lacks transparency.

### 6. Conclusion

This paper introduced a neurosymbolic approach to VQA that combines neural perception with symbolic reasoning. By leveraging pre-trained models in a zero-shot setting, we avoid task-specific retraining while maintaining flexibility across different types of questions and visual scenes. The integration of ASP enables structured reasoning over symbolic representations, improving both interpretability and error traceability. Our experiments on the GQA dataset demonstrated the feasibility of this approach, and we illustrated how existing tools can provide step-by-step explanations of the reasoning process. Beyond explaining its own answers, we can also produce derivation traces for answers given by more opaque models by incorporating their predictions into our reasoning process.

There are several directions for future research. While our approach benefits from a zero-shot setup, incorporating task-specific fine-tuning could enhance the quality of scene graphs and question parsing. Expanding evaluations to additional VQA datasets would further assess the framework's adaptability to different domains. Furthermore, as stronger LLMs and VLMs become available, we expect further improvements in accuracy, helping to bridge the current performance gap with existing neurosymbolic approaches by improving both perception and question interpretation.

## Acknowledgments

This work was supported by the Bosch Center for AI.

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
