# OpenReview forum: "Explainable Zero-Shot Visual Question Answering via Logic-Based Reasoning"
_nesyconf.org/NeSy/2025/Conference — NeSy 2025 Poster_

### Official Review · Reviewer_7X2H · 2025-03-31
**Not enough scientific contribution**

**Rating:** 3
**Confidence:** 3

**Review:**

The paper presents a Neurosymbolic method for Visual Question Answering (VQA). The method is a pipeline where a scene graph is extracted with a Vision Language Model (VLM), the question is parsed and translated in an Answer Set Programming (ASP) problem with a Large Language Model (LLM), the answer to the original question is performed through Answer Set Programming by considering the scene graph and the parsed question. This is a hot and interesting topic in the Computer Vision community, however, the paper is just a composition of well-known techniques (LLMs, VLMs and the works of Eiter about VQA with ASP). There are no underlying/explicit research questions and both the explainability and the property of being zero-shot is inherently due to the use of ASP. Therefore, a demo section of some AI conference would be a better venue for this kind of work.

Other concern:
- In Section 4.2.2, the authors claim that only the concepts inside the questions are retained in the graph. While this solve many issues about the complexity of the scene graph, this could create problems with more open questions where the answer cannot be deduced by only using concepts inside the question. For example, the question "Tell me the class of the green objects" can involve several concepts such as, meadow and bush, not listed int the question.
- The above point would have been an interesting research question.
- The acronym GS-VQA is used from page 2 but explained only at page 4.

**Anonymity:**

Disclose identity

---

### Official Review · Reviewer_JRDj · 2025-04-03
**The work introduces a framework for explainable visual question answering. Although the performances are low, the proposed methods and pipeline remain interesting and relevant for the community.**

**Rating:** 7
**Confidence:** 3

**Review:**

The paper proposes a hybrid neuro-symbolic pipeline for learning-free Visual Question Answering, referred to as Grounded-Scene Visual Question Answering (GS-VQA). Logic-based reasoning within this pipeline is implemented using Answer-Set Programming (ASP).

The evaluation is performed using the GQA dataset, a widely recognized and adopted benchmark for visual question-answering tasks characterized by its challenges with respect to reasoning requirements.

The proposed pipeline is composed of three main components:

Language Module: Employs the GPT-4 model, which parses natural-language questions and constructs structured representations translated into ASP facts. Importantly, this transformer-based model is neither trained nor fine-tuned. Instead, it leverages a few-shot prompting approach.

Vision Module: Utilizes OWL-ViT for object detection and CLIP for concept classification, enabling a question-driven partial scene-graph extraction. This targeted extraction approach addresses the complexity of the GQA dataset by focusing only on scene components relevant to the given question.

Reasoning Module: Integrates outputs from both the Language and Vision modules, translating them into ASP facts and solving them through an ASP solver. This allows for structured and interpretable logic-based inference.

The system ensures high transparency by tracing reasoning processes through an ASP-based explanation tool (xclingo). This enables the interpretability of correct answers and improves diagnostic capabilities for incorrect responses.
Additionally, the authors show the capability of their system to generate explanations for responses produced by other opaque VQA models.

Despite its methodological merits, the empirical results indicate that the proposed system's accuracy significantly trails behind the leading zero-shot VQA methods, performing substantially worse.

Structurally, the paper is clear and well written. The pipeline and methods are well-explained. Figures and examples (i.e. prompts for the language module) contribute well to the readability and understanding of the paper.

Pros:
- Clear and transparent reasoning via ASP.
- Tracing of the reasoning process via derivation trees.
- Ability to generate explanations for other opaque, completely neural models.
- Very well written and easy to follow.
- Code is provided.

Cons:
- Heavily relies on pre-trained models.
- Concerns regarding the computational complexity.
- Accuracy is not very high

This paper is interesting and relevant to the community, therefore I recommend acceptance.

**Anonymity:**

Remain anonymous

---

### Official Review · Reviewer_npJ5 · 2025-04-14
**A clearly written paper with a neat combination of existing methods**

**Rating:** 7
**Confidence:** 4

**Review:**

The paper provides an interpretable zero shot method for the task of visual question answering. It uses existing foundation models that have been pretrained to solve the task of VQA in an interpretable manner using them to generate function programs from the questions and scene graphs for the images. The paper also provides how their method can be used for explainability of opaque systems as well.

The paper is clearly written, provides a neat and simple idea for zero-shot VQA. I elaborate on the strengths and weaknesses of the work below:

**Strengths**:

1. The paper provides a simple combination of existing methodologies to provide a zero shot solution for the task of VQA.
2. The paper is clearly written and explains the methodology in good detail.
3. The paper positions itself well with respect to the existing work.

**Weaknesses**:

1. The paper currently uses existing models such as GPT-4 (released in 2024) and uses the dataset GQA (released in 2019). It is very possible that GQA dataset was used in the training of GPT-4. Thus, I am skeptical if the method would adapt so well on novel questions that are not available in the training of the LLM used. I can still see how the paper offers a clever use of foundation models to solve the VQA task without additional training in an interpretable way.
2. The evaluation of the paper is not very thorough. It evaluates the full pipeline and the language model but does not evaluate the vision module.
3. The paper talks about uncertainty in the vision module and includes that as part of the reasoning through ASP but ignores the uncertainty due to the language module.

**Minor Comments**:

In Section 2, the paper says that “prior work assumed pre-constructed symbolic representations, omitting the parsing step”. This is not entirely true as Mao et al. trains a language module that parses the questions into programs in their own DSL. Admittedly, Mao et al.’s approach is not zero-shot.

**Justification for the score**:

Given the above strengths and weaknesses of the paper, I consider the paper to be providing value through a neat methodology to use pre-trained neural modules to solve the task of VQA.

**Anonymity:**

Remain anonymous